# Computer-Aided and AILDE Approaches to Design Novel 4-Hydroxyphenylpyruvate Dioxygenase Inhibitors

**DOI:** 10.3390/ijms23147822

**Published:** 2022-07-15

**Authors:** Juan Shi, Shuang Gao, Jia-Yu Wang, Tong Ye, Ming-Li Yue, Ying Fu, Fei Ye

**Affiliations:** Department of Applied Chemistry, College of Arts and Sciences, Northeast Agricultural University, Harbin 150030, China; shijuan1233@163.com (J.S.); gaoshuang@neau.edu.cn (S.G.); wangjiayu9819@163.com (J.-Y.W.); yetong0812@163.com (T.Y.); yueml@neau.edu.cn (M.-L.Y.)

**Keywords:** HPPD, Topomer CoMFA, molecular docking, molecular dynamics, AILDE

## Abstract

4-Hydroxyphenylpyruvate dioxygenase (HPPD) is a pivotal enzyme in tocopherol and plastoquinone synthesis and a potential target for novel herbicides. Thirty-five pyridine derivatives were selected to establish a Topomer comparative molecular field analysis (Topomer CoMFA) model to obtain correlation information between HPPD inhibitory activity and the molecular structure. A credible and predictive Topomer CoMFA model was established by “split in two R-groups” cutting methods and fragment combinations (*q*^2^ = 0.703, *r*^2^ = 0.957, *ONC* = 6). The established model was used to screen out more active compounds and was optimized through the auto in silico ligand directing evolution (AILDE) platform to obtain potential HPPD inhibitors. Twenty-two new compounds with theoretically good HPPD inhibition were obtained by combining the high-activity contribution substituents in the existing molecules with the R-group search via Topomer search. Molecular docking results revealed that most of the 22 fresh compounds could form stable π-π interactions. The absorption, distribution, metabolism, excretion and toxicity (ADMET) prediction and drug-like properties made 9 compounds potential HPPD inhibitors. Molecular dynamics simulation indicated that Compounds Y12 and Y14 showed good root mean square deviation (RMSD) and root mean square fluctuation (RMSF) values and stability. According to the AILDE online verification, 5 new compounds with potential HPPD inhibition were discovered as HPPD inhibitor candidates. This study provides beneficial insights for subsequent HPPD inhibitor design.

## 1. Introduction

4-Hydroxyphenylpyruvate dioxygenase (HPPD) is a catalytic enzyme in the synthesis of plastoquinone (PQ) and tocopherol in organisms that converts 4-hydroxyphenylpyruvate (HPPA) to homogentisic acid (HGA) [1,2,3]. Phytoene will accumulate once the production of PQ is affected. If the HPPD inhibitor interferes with the conversion of HPPA to HGA, it will lead tissue necrosis and albinism symptoms and even plant death [4,5]. Plants in the sun may eventually suffer death with albino symptoms if HPPD inhibitors interfere with the transformation of HPPA to HGA. As a consequence, HPPD becomes a critical latent target for developing novel herbicides [6,7,8,9]. HPPD based herbicides play a vital role in rice production by inhibiting the photosynthesis of weeds [10,11]. However, contradictions, such as crop selectivity, weed resistance, herbicide residues, and cost in the development of new herbicides, appear gradually, so it is urgent to explore new green HPPD inhibitors to address these challenges [12,13].

Computer technology promotes drug development and provides theoretical guidance for drug design. Topomer comparative molecular field analysis (Topomer CoMFA), a combination of the initial “Topomer” method and CoMFA technology with 3D-quantitative structure-activity relationship (3D-QSAR) technology and autocomplete regression analysis can be used to predict the physicochemical properties or biological activity of compounds and screen the database [14,15]. Topomer CoMFA model has been widely used in drug design, such as for Alzheimer’s disease, human immunodeficiency virus (HIV), hypercholesterolemia, and breast, lung or renal cell carcinoma [16,17,18,19]. The most attractive feature is that Topomer CoMFA provides a relatively objective model because of the wholly automatic process [20,21]. In addition, it facilitates the development of emerging target enzyme inhibitors. The models based on 37 derivatives of 2-phenylquinazolin-4-one predicted novel quinazolinone derivatives as potent tankyrase inhibitors [22,23,24]. A Topomer CoMFA model was established based on sulfonylurea herbicides, by which 36 new potential inhibitors were obtained by filtration in the ZINC database [25]. Novel antibacterial agents against phytophthora capsici and cucumber peronospora were developed by the Topomer CoMFA model based on carboxylic acid amide fungicides [26]. Fourteen potential human HPPD (*h*HPPD) inhibitors were identified by Topomer CoMFA model screening [27].

Molecular dynamics (MD) simulation is useful for studying protein motion. Drug binding and molecular recognition can be studied by MD simulations [28,29]. Binding free energy identifies the stability of the binding of major residues in protein-ligand interactions, and the molecular mechanical potential energy and solvation free energy during the binding process of the enzyme and the inhibitor have been calculated [30,31]. It revealed that hesperidin, remdesivir, quercetin and sulabiroin-A may be potential natural inhibitors of severe acute respiratory syndrome coronavirus 2 (SARS-CoV-2) [32]. Two potential triketone herbicides have been compared with commercial mesotrione and (2-(aryloxyacetyl) cyclohexane-1,3-dione) by MD simulations, and the screening results are helpful to obtain HPPD inhibitors [33]. The calculation results are more intuitively represented by root mean square deviation (RMSD) and root mean square fluctuation (RMSF) [34].

Hit-to-lead (H2L) is employed to optimize the structure of the compounds, which has been increasingly studied in medicinal chemistry. It has been used against hepatitis B virus (HBV) polycyclic pyridone drugs [35]. In research on antimalarial drugs, two potential drugs were designed via the H_2_L method, and the obtained compound activities were verified [36]. Auto in silico ligand directing evolution (AILDE, http://chemyang.ccnu.edu.cn/ccb/server/AILDE/ accessed on 10 February 2021) technology can rapidly identify drug leads in close chemical space [37]. AILDE greatly improves the discovery and synthesis efficiency of potential inhibitors, by which it chemically modifies the molecular fragments of the hit compounds [38].

In this study, a Topomer CoMFA model was built based on 35 pyridine HPPD inhibitors, and potential inhibitors were formed through virtual screening of the more active R-groups in the Bailingwei database (approximately 50,000 fragments). Molecular docking studies further elucidated the interaction between the ligand and receptor. The absorption, distribution, metabolism, excretion and toxicity (ADMET) of the obtained compounds were calculated. The compounds with the best MD simulation verification were submitted to the AILDE server to promote the hit compounds. The filter strategy is shown in Figure 1.

## 2. Results and Discussion

### 2.1. Topomer CoMFA Analysis

The Topomer CoMFA model statistical results are shown in Table 1.

The large *q*^2^ and *r*^2^ (*q*^2^ = 0.703, *r*^2^ = 0.957) indicated that the prediction was statistically significant. Linear regression plots of experimental and predicted data of HPPD inhibition are shown in Figure 2, showing that the experimental and predicted values were uniformly distributed near the 45° line, which indicated excellent prediction ability. This feature was common in multiple models with screening and prediction capabilities [39,40,41].

The steric and electrostatic fields are displayed as contour maps. Figure 3a,b show the steric contour maps of the R^1^ and R^2^ groups. The green profile enhanced the herbicidal activity by bulky substituents, while the yellow profile enhanced the inhibitory effect by small substituents. A green outline with a large overlap with the plane of the 2-, 3- and 4-positions of the piperidinone of ZD-12 (IC_50_ = 0.325 μM) indicated that the activity can be increased by selecting bulkier substituents at these sites (Figure 3a). As displayed in Figure 3b, a large green outline near the 3-position was also found in the R^2^ group. For example, compounds ZD-8 (IC_50_ = 0.283 μM) and ZD-9 (IC_50_ = 0.261 μM) exhibited good activities because of mesyl substituents.

The electrostatic fields are shown in red and blue in the contour maps (Figure 3c,d). For increased weeding activity, red is suitable for negatively charged substituents, and blue isolines are suitable for positively charged substituents. As revealed in Figure 3c, position 4 of the pyridine ring showed a large red outline, and this position should be substituted by a negative charge. The blue outline near the N atom indicated that selecting a positively charged substituent at the N atom site would increase the activity. In Figure 3d, there is a large red outline at the 2 and 3 positions of the phenyl group, indicating that negative charge substitution at this position would increase the activity. For example, ZD-24 (IC_50_ = 1.202 μM) was less active than ZD-27 (IC_50_ = 0.425 μM) due to the methyl being replaced by -Br at the 2-position. For example, the activity of ZD-3 (IC_50_ = 1.365 μM) was worse than ZD-4 (IC_50_ = 0.998 μM) due to methyl being replaced by -CF_3_ at the 3-position, and the activity of ZD-29 (IC_50_ = 1.589 μM) was worse than ZD-26 (IC_50_ = 0.998 μM) due to the methyl being replaced by -Cl at the 3-position. A similar effect was observed between compounds ZD-33 (IC_50_ = 7.656 μM) and ZD-32 (IC_50_ = 1.419 μM).

### 2.2. Topomer Search

Topomer search was employed to screen similar structures or high contribution substituents with HPPD inhibitory activity. The R^2^-group model was developed to screen for approximately 50,000 fragments. The hit substituents were ranked according to their value of contribution to activity. Ninety-five new compounds were generated with a contribution value greater than 0.2 as the standard. Then, 64 candidates were obtained by selecting the configuration with the highest score in molecular docking. Finally, 22 molecules were obtained based on the existence of metal coordination bonds. The molecular structures and docking results are shown in Table 2.

### 2.3. Molecular Docking Analysis

The CDOCKER program was used to explore the interaction between HPPD (PDB ID: 6JX9) protein residues and ligands [42]. The results were verified by comparing the RMSD between the original ligand and the same interaction site as the original ligand. As shown in Figure 4, the ligand after redocking of the original ligand (red) completely overlapped with the ligand in the complex (cyan) and had exactly the same π-π stacked interactions of Phe381 and Phe424, indicating that the selected protein can be used as a docking model.

Subsequently, all the newly designed molecules and the most active ZD-9 were docked into 6JX9, and the -CDOCKER energy is shown in Table 2. The -CDOCKER energies were all above that of the native ligand (51.69 kcal/mol), which confirmed that the designed compounds exhibited better docking results coincidence than the native ligand. The -CDOCKER energy (65.51 kcal/mol) of Y12 was greater than that of the native ligand. The π-π stacked interactions between Y12 and Phe392 may be the reason for the -CDOCKER energies of Y12 compared with the native ligand. The most active compound, ZD-9, was selected as the template for detailed description. Two compounds, Y12 and Y14, with the best -CDOCKER energy were selected to analyze the binding mode at the active pocket.

As shown in Figure 5a, ZD-9 was fully embedded into the active pocket. The hydrogen atom of the imino formed hydrogen bonds with the oxygen atoms of Ser267; the two carbonyls produced metal coordination bonded with cobalt ions; and the benzene rings formed π-π interactions with Phe381 and Phe424. This was consistent with the docking results reported earlier [43]. Figure 5b showed that the π-π interaction of the pyridine ring of Compound Y12 was similar to that of ZD-9. Moreover, Phe381 also formed a π-π interaction with the furan ring of Compound Y12, and Gln379 formed hydrogen bonds with carbonyl outside the pyridine ring, which made Compound Y12 binding more stable than ZD-9. Compound Y14 was also well inserted into the active site of the protein, as shown in Figure 5c. It also interacted with amino acids Phe381 and Phe424, similar to ZD-9, and the necessary metal coordination bonds and hydrogen bonds also exist.

### 2.4. ADMET Prediction

Then, the 22 newly designed compounds were subjected to ADME analysis. The molecules that exist in the 99% confidence interval of the blood–brain barrier permeability model and the 99% confidence interval of the human intestinal absorption model were selected as the hit compounds, and 18 of them showed ADME properties in the acceptable range (Figure 6). The bayesian score of compounds in the hepatotoxic were less than −0.41 (Table 3). Therefore, these newly designed molecules had excellent physical and chemical properties, low toxicity and good crop protection.

The toxicity prediction (extensible model of ADME) results indicated that the reference compound ZD-9 is non-mutagenic, noncarcinogenic and degradable. Only nine of the newly designed compounds met the criteria of non-mutagenicity, non-carcinogenicity and degradability (Table 4).

### 2.5. MD Simulations

To further determine the accuracy of the docking procedure, the native ligand and the redocked ligand were simulated by MD simulation. The active pocket of the 5 Å residue around the ligand stabilizes after 10 ns, and the redocked ligand was below the native ligand, indicating that the redocking procedure will be more stable (Figure 7a). The skeleton Cα atoms of redocked ligand and native ligand fluctuate at about the same trend after 7.5 ns (Figure 7b). The heavy atoms of the native ligand-receptor were 2.5 ns ahead of the redocked ligand-acceptor (Figure 7c). The native ligand and the redocked ligand proved the docking procedure was correct under similar fluctuations and stability.

To further determine the accuracy of the hit, 9 compounds screened by ADMET were simulated by MD simulation to determine whether the binding of the HPPD-ligand complex was stable. The overall stability of the system was evaluated by monitoring the RMSD of skeleton atoms. Skeleton Cα atoms of proteins, active pockets of 5 Å residues around ligands and heavy atoms of ligands-receptor were simulated in 50 ns. The side chain flexibility was generally higher than that of the backbone atoms; therefore, the RMSD of the main chain is a crucial indicator of system stability.

Figure 8a showed that the residues within 5 Å around the protein active pocket ligand stabilized after 20 ns, and then, the RMSD value remained unchanged. In Figure 8b, the HPPD-Y12 and HPPD-Y14 complexes fluctuated in a small range at the beginning and gradually reached equilibrium after 5 ns of simulation. The RMSD values of the remaining seven HPPD complexes were stable after 10 ns. The stable RMSD values of complexes Y12 (1.0 Å) and Y14 (0.5 Å) were significantly lower than those of the other seven compounds, indicating that these two complexes were more stable. The heavy atoms of the ligands-receptor tended to balance after 10 ns, and the RMSD value remained stable (Figure 8c).

A lower RMSD of the docking complexes was a good indicator of system stability. The structural fluctuations of the HPPD-Y12 and HPPD-Y14 systems were relatively lower than those of the other molecules. These results suggested that Y12 and Y14 exhibited relatively favorable binding affinity with HPPD.

In addition, the results of RMSF are shown in Figure 9. The peak represents the region where the protein fluctuates the most during the simulation. Due to hydrogen bonding between Asn282, Ser267 and ligands, structural flexibility was significantly increased in the residue index of 250–300, and the residue index of 350–400 was the π-π stacked interactions on account of Phe381 (Figure 9a). It was generally observed that the tail (N-terminal and C-terminal) fluctuated more than other parts of the protein. Secondary structural elements such as alpha helical and beta strands are generally less volatile than unstructured parts of proteins than loop regions. This phenomenon was also seen in (Figure 9b,c). All compounds characterized local changes in the protein chain and simulated fluctuations in RMSF values corresponding to the terminal portion which were higher than the intermediate portion.

### 2.6. AILDE Optimized Hit Compounds and Physicochemical Properties

The AILDE method was the ligand substitution scanning mutagenesis calculation method, one of the strategies used in H_2_L optimization in agricultural chemical design [44]. Compound Y12 was submitted to the online server for AILDE verification because it exhibited the best molecular docking and MD simulation verification results. The activities of the compounds were input into the AILDE server, and computation substituent optimization was performed by replacing the hydrogen number in the structure (Figure 10a). The results of the heat map in Figure 10b showed that the redder the compound was, the better the activity was. For example, if the hydrogen at position 5871 was replaced by chlorine, the hydrogen at position 5897 was replaced by chlorine or amino, or if the hydrogen at positions 5898 and 5903 were replaced by bromine, more active compounds would be obtained. The overall results based on the corresponding substituents are illustrated in the histogram in Figure 10c.

The greatest potential transformation was shown at positions 5871, 5897, 5898, and 5903. Then, five compounds were obtained after screening (Table 5). If the substitution of the hydrogen atom at position 5897 was by an amino group, the Δ*G* value was the smallest; thus, the compound had priority. In summary, the use of AILDE will improve the efficiency of initial drug discovery, which also provides favorable guidance for subsequent synthesis work.

The amount of HBA, HBD and AR in the compound is positively correlated with biological activity [45,46]. It is worth noting that the number of HBA and HBD of S3 is slightly higher than that of Y12 (Table 6). The electronegativities of the five compounds obtained are very similar, and their predicted p*K*a values are less than 6.0, which are even lower than that of Y12 without optimization. Weak acidity and low Log *p* are conducive to the spread and absorption of plants. Based on the physical and chemical properties, the optimized process is beneficial for obtaining new HPPD inhibitors.

## 3. Methods and Materials

### 3.1. Information Collection

The structures and bioactivities of the 35 pyridine derivatives employed in this experiment were derived from our laboratory [43]. These pyridine derivatives are different in aromatic subunit. The collection of derived compounds contained 28 compounds as training sets and 7 compounds as test sets. Topomer CoMFA model was constructed with these compounds, and the predictive ability of the model was tested. The bioactivity and chemical structure are reported using the pIC_50_ data in Table 7.

The Sketch module in SYBYL was used to construct molecular structures. All compounds were perfected using a Tripos force field and gradient descent with an energy charge of 0.005 kcal/mol. Gasteiger–Huckel adds partial charges to all compounds. The maximum iteration coefficient was selected as 1000, and the other parameters default to SYBYL. Energy minimization was optimized for all molecules. The minimum energy conformation of the best active compound ZD-9 (6-hydroxy-5-(4-(methylsulfonyl)-2-nitrobenzoyl)-2,3-dihydropyridin-4(1H)-one) was selected as the template compound.

### 3.2. Topomer CoMFA Modeling Construction and Virtual Screening

The template compound ZD-9 was split into two R-groups (Figure 11). Other compounds in the training set were dissected automatically in ZD-9 segmentation manner. Unrecognized molecules needed to be cut manually. After cutting, the fragment conformation was adjusted according to empirical rules to generate the Topomer CoMFA model.

Highly active R-groups were screened from the Bailingwei (2012) database (approximately 50,000 fragments) using Topomer Search. Topomer Search-Details of Distance (TOPDIST) was selected at 185 to identify the degree of binding. Twenty-two novel pyridine derivatives were obtained, and their activities were predicted by the established Topomer CoMFA model.

### 3.3. Molecular Docking

Molecular docking studies were performed using CDOCKER of Discovery Studio (DS) (Biovia Inc., San Diego, CA, USA, 2020). The pose cluster radius was set to 0.5 in this program to ensure that docking ideas were as diverse as possible. The complex structure of HPPD (PDB ID: 6JX9) was obtained from the RCSB protein database [42]. Co-crystalline Y17107 was extracted from the enzyme structure, hydrogen was added, and all the heteroatoms and water molecules were removed. After replenishing all the missing amino acids, the structure of HPPD was protonated by the CHARMm force field, and all amino acid side chains were optimized.

### 3.4. ADMET Prediction

The top 22 compounds for the molecular docking score were predicted in terms of absorption, distribution, metabolism, excretion and toxicity (ADMET) with DS. Four pharmacokinetic parameters were calculated, including water solubility, cytochrome P450 (CYP450) binding and hepatotoxicity [43]. Five parameters were selected for toxicity prediction: oxygen biodegradability, carcinogenicity, ames mutagenicity, developmental toxicity potential and skin irritation screening.

### 3.5. MD Simulation

To predict the stability of HPPD protein binding to inhibitors, Amber 16 software was selected for the MD simulation. The side chain models of the cobalt coordination spheres of His308, His226 and Glu394 were used to create a builder module in the center of the metal by selecting the ff14SB field. The resulting structures were immersed in TIP3P water at a distance of 10 Å around the complexes, and an appropriate amount of counter ions were appended to the system to neutralize charges. The minimum process used a 2500-step conjugate gradient and 2500-step maximum descent algorithm. The system was gradually heated to 298 K by an isovolumic-isothermal, and equilibrium was achieved by simulating 1 ns in the isobaric-isothermal ensemble simulation. Finally, each system executed the PMEMD program unconstrained in the NTP integration for 50 ns with a time step of 2 fs. The root mean square deviation (RMSD) was used to evaluate the stability of the system.

### 3.6. AILDE and Physicochemical Properties

AILDE automatically performs calculations of substitution, energy minimization, and binding affinity assessments. All hydrogen was removed from the docking ligand-protein complex, and the compound was submitted to the online server. The physicochemical properties of molecules were calculated by the “Small Molecules” of DS. The following parameters were predicted in the 2D topology descriptor: Log *p*, p*K*a, molecular weight (*M*_W_), hydrogen bond acceptors (HBAs), hydrogen bond donors (HBDs), aromatic bonds (ARs) and surface area (SA). The electronegativity of the compounds was calculated using the “Connolly” of Gasteiger–Huckel of SYBYL software.

## 4. Conclusions

In this study, a reliable Topomer CoMFA model was established based on 35 pyridine HPPD inhibition herbicides. Twenty-two new compounds were designed via Topomer search according to the Topomer CoMFA model. The molecular docking results indicated that the ligands formed hydrogen bonds with Ser267 and π-π interactions with Phe381 and Phe424 at the active site. The 22 newly designed compounds were filtered through ADMET prediction, and finally, 9 compounds were obtained. MD simulations confirmed that Compounds Y12 and Y14 bear potential HPPD inhibition. Y12 was submitted to the AILDE platform, and 5 potential inhibitors were found in the activity-improved matrix. The established screening procedure is of great significance for the design of novel HPPD based herbicides.

## Figures and Tables

**Figure 1 ijms-23-07822-f001:**
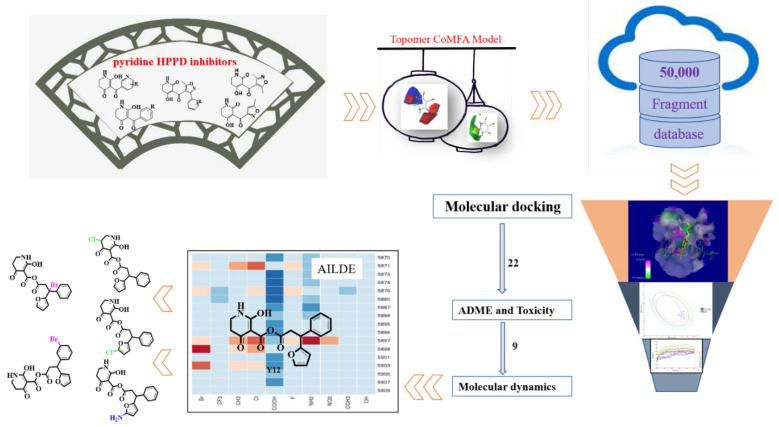
The screening workflow applied to design novel HPPD inhibitors.

**Figure 2 ijms-23-07822-f002:**
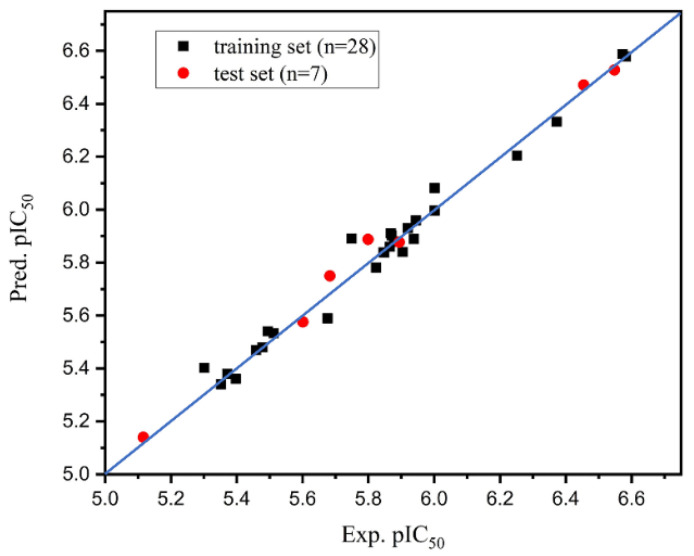
Plot of the experimental and predicted values for training and test set compounds using Topomer CoMFA model.

**Figure 3 ijms-23-07822-f003:**
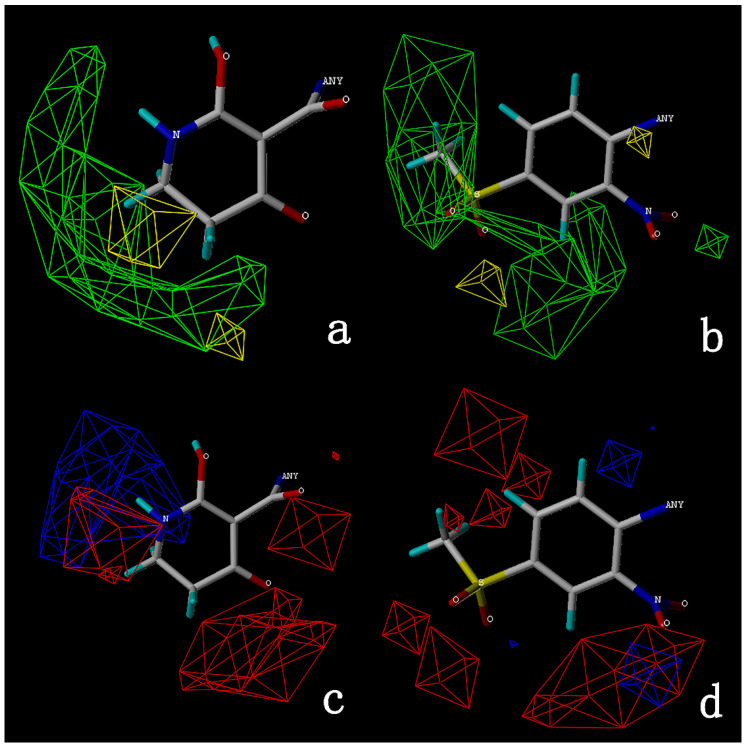
Topomer CoMFA steric (**a**,**b**) and electrostatic (**c**,**d**) contour maps based on the R^1^ and R^2^ groups of the most active ZD-9.

**Figure 4 ijms-23-07822-f004:**
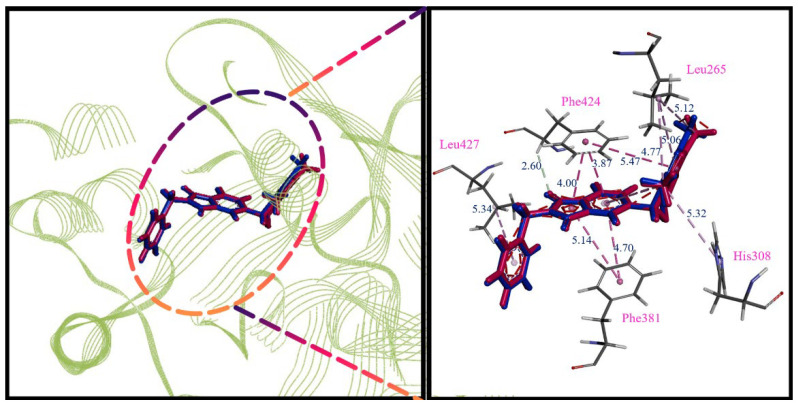
Alignment of the redocked ligands with the native ligand in the crystallographic complex. Redocked ligand is cyan, the native ligand is red. Green is the distance of the interaction (Å).

**Figure 5 ijms-23-07822-f005:**
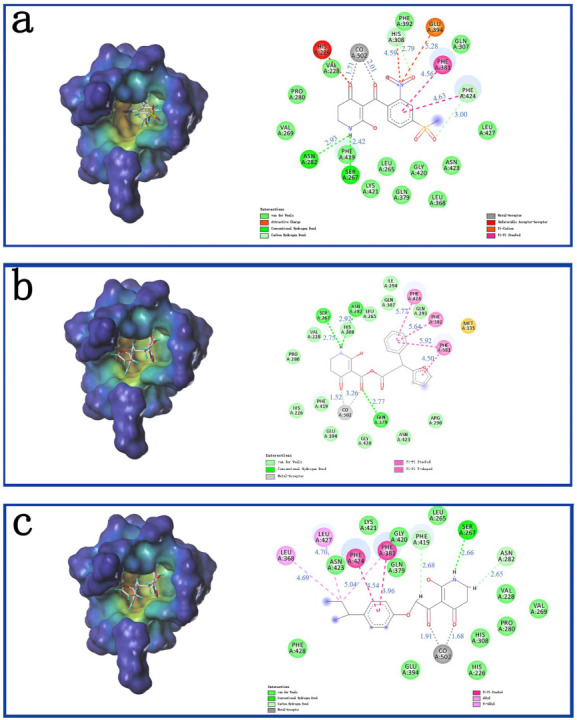
The receptor-ligand interaction of ZD-9 (**a**), Y12 (**b**) and Y14 (**c**) in the HPPD active site. Blue is the distance of the interaction (Å).

**Figure 6 ijms-23-07822-f006:**
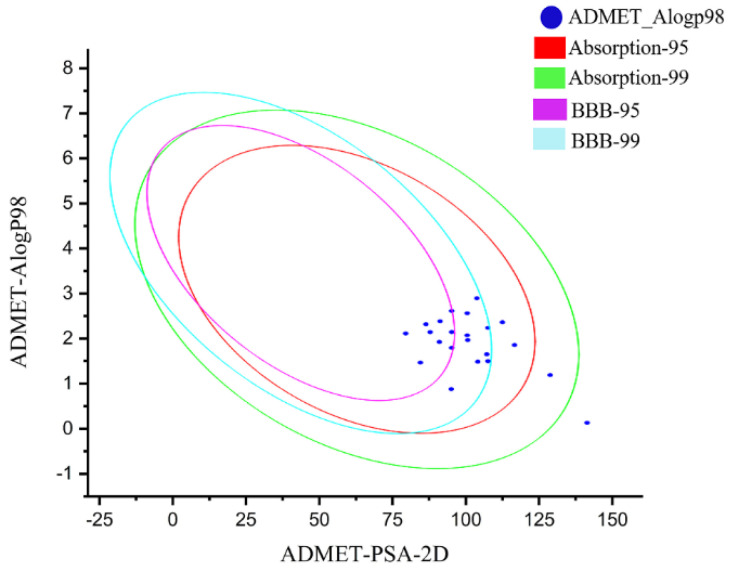
ADMET prediction for all designed compounds.

**Figure 7 ijms-23-07822-f007:**
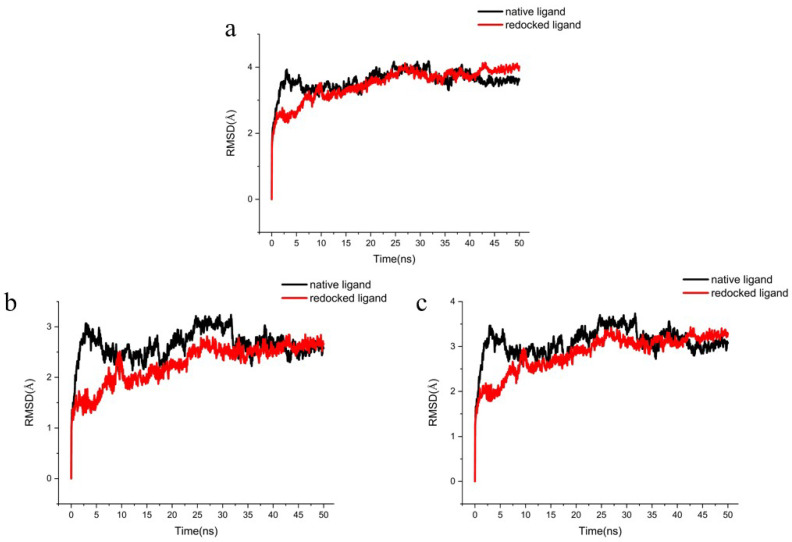
The RMSD trajectories of the redocked ligands with the native ligand during 50 ns simulations. (**a**) The protein active pocket with residues of 5 Å around ligand, (**b**) backbone Cα atoms and (**c**) heavy atoms of ligand-receptor.

**Figure 8 ijms-23-07822-f008:**
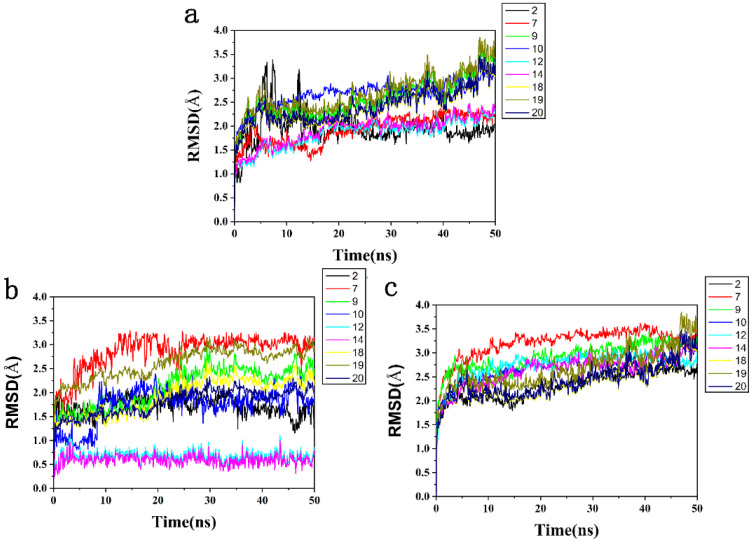
The RMSD trajectories of all systems during 50 ns simulations. (**a**) The protein active pocket with residues of 5 Å around ligand, (**b**) backbone Cα atoms and (**c**) heavy atoms of ligand-receptor.

**Figure 9 ijms-23-07822-f009:**
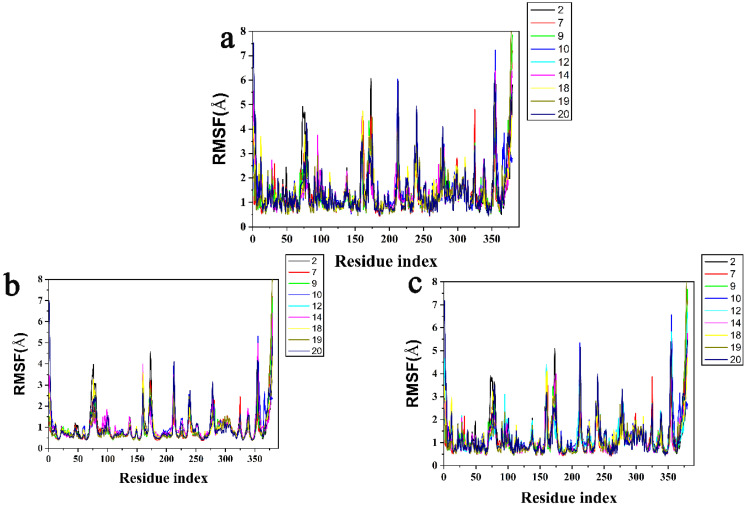
Protein RMSF for (**a**) the protein active pocket with residues of 5 Å around (**b**) Cα atoms and (**c**) heavy atoms.

**Figure 10 ijms-23-07822-f010:**
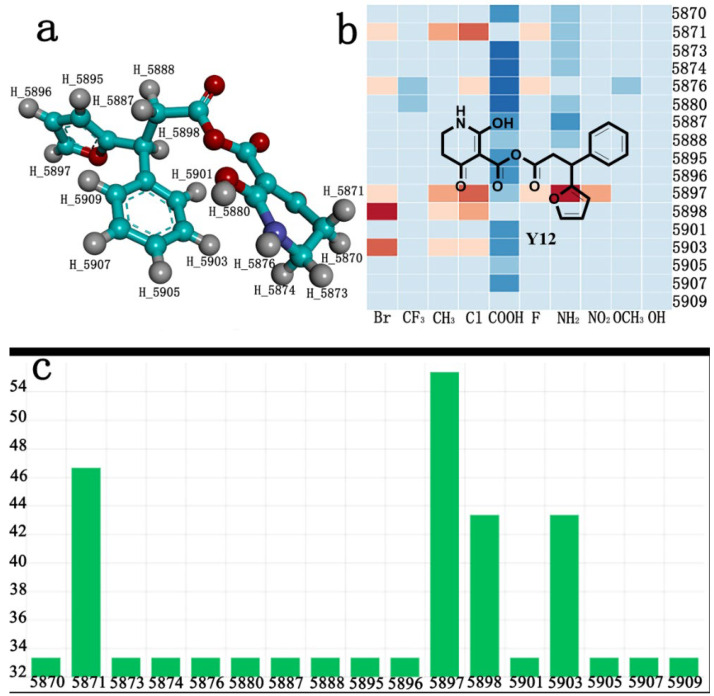
(**a**) Hydrogen serial numbers; (**b**) Heat map showing the relationship matrix between the substituent positions and substituents (darker red colors indicate compounds with better activities); (**c**) Histogram illustrating the overall result based on substituent positions, which helps to elucidate which positions possess the most potential for substitution (X: Hydrogen number, Y: Score).

**Figure 11 ijms-23-07822-f011:**
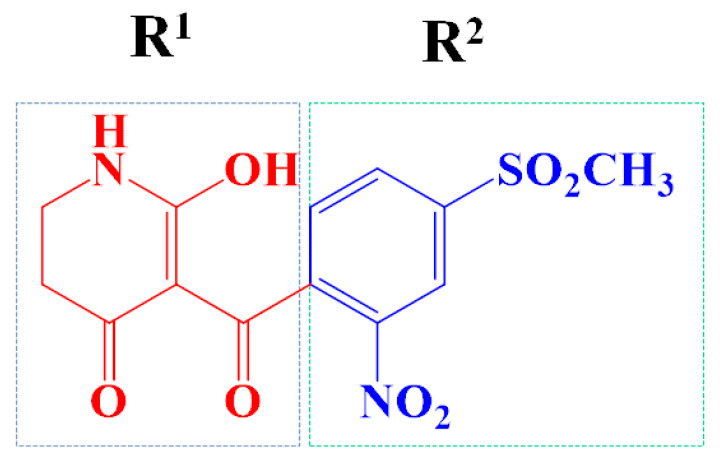
Splitting scheme used to define the two fragments (R^1^—red and R^2^—blue) for developing Topomer CoMFA model of HPPD inhibitors.

**Table 1 ijms-23-07822-t001:** Statistical Results of the Topomer CoMFA.

Cutting Method	*q* ^2^	*r* ^2^	*N*	*F*	*SEE*	*Intercept*
“split in two R-groups”	0.703	0.957	6	95.338	0.046	4.98

**Table 2 ijms-23-07822-t002:** Structures and -CDOCKER Energy (kcal/mol) of novel designed compounds.

Comp.	Structure	-CDOCKER Energy	Comp.	Structure	-CDOCKER Energy
ZD-9	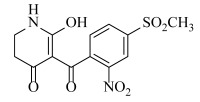	53.62	Native ligand	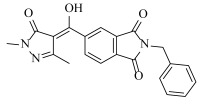	51.69
Y1	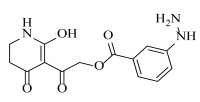	62.76	Y12	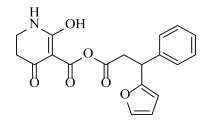	65.51
Y2	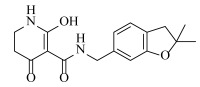	51.79	Y13	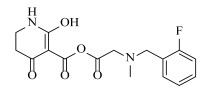	61.75
Y3	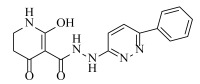	51.66	Y14	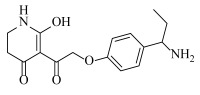	69.09
Y4	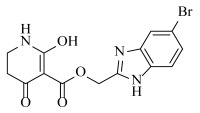	63.96	Y15	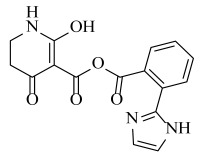	59.92
Y5	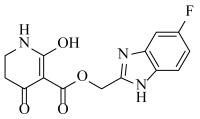	54.27	Y16	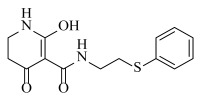	59.81
Y6	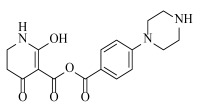	59.48	Y17	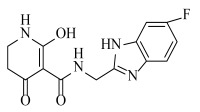	58.60
Y7	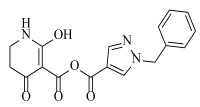	65.57	Y18	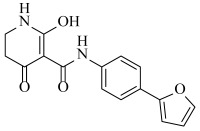	54.28
Y8	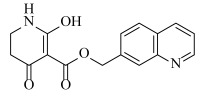	53.29	Y19	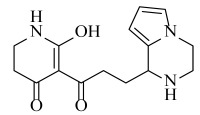	59.89
Y9	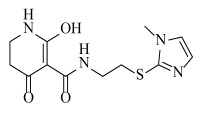	52.83	Y20	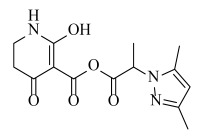	55.52
Y10	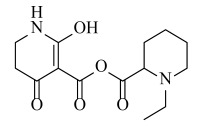	58.18	Y21	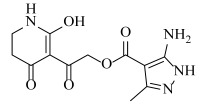	54.60
Y11	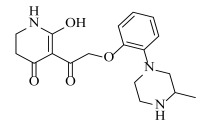	65.12	Y22	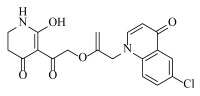	60.43

**Table 3 ijms-23-07822-t003:** The ADME predictions of compounds.

Comp.	Solubility Level	Hepatotoxic	CYP2D6_ Applicability
Y1	3	−2.52	13.33
Y2	3	−0.71	18.64
Y3	3	−0.99	15.80
Y4	3	−1.57	14.34
Y5	3	−0.46	13.20
Y6	3	−0.89	12.68
Y7	3	−2.85	15.91
Y8	3	−2.03	14.99
Y9	3	−0.43	17.18
Y10	3	−4.56	11.17
Y11	4	−3.20	12.81
Y12	3	−1.68	11.07
Y13	3	−2.06	13.78
Y14	3	−3.28	13.22
Y15	3	0.84	15.14
Y16	4	−0.73	13.56
Y17	3	−2.25	14.67
Y18	3	−3.45	13.58
Y19	4	−1.24	15.39
Y20	4	−0.09	19.39
Y21	4	−1.01	18.93
Y22	3	−3.68	13.27

Solubility Level: Categorical solubility level. 2: Yes, low; 3: Yes, good; 4: Yes, great. Hepatotoxic: <−0.41: nontoxic; >−0.41: toxic.

**Table 4 ijms-23-07822-t004:** Toxicity prediction of the new compounds and reference compounds.

Comp.	Degradability	Mutagenicity	Carcinogenicity
ZD-9	√	×	×
Y1	×	×	×
Y2	√	×	×
Y3	×	√	×
Y4	×	×	√
Y5	×	√	×
Y6	×	×	×
Y7	√	×	×
Y8	×	√	×
Y9	√	×	×
Y10	√	×	×
Y11	×	×	×
Y12	√	×	×
Y13	×	×	×
Y14	√	×	×
Y15	√	×	×
Y16	×	×	×
Y17	×	√	×
Y18	√	×	×
Y19	√	×	×
Y20	√	×	×
Y21	×	×	×
Y22	×	×	×

**Table 5 ijms-23-07822-t005:** Screening result for 5 compounds.

Comp.	S1	S2	S3	S4	S5
Structure	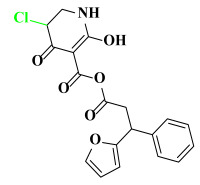	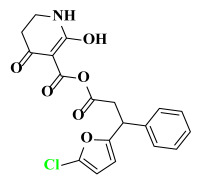	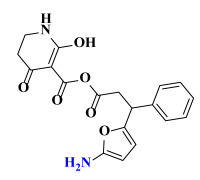	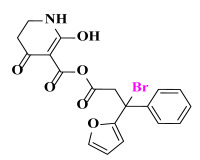	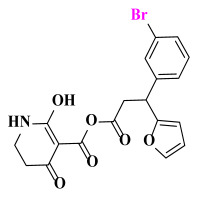
Hydrogen Number	5871	5897	5897	5898	5903
Δ*G* (kcal/mol)	−1.23	−0.98	−1.87	−1.53	−1.23
Δ*H* (kcal/mol)	−1.61	−1.27	−2.24	−1.72	−1.25
*−T*Δ*S* (kcal/mol)	0.38	0.29	0.37	0.19	0.03

**Table 6 ijms-23-07822-t006:** Screening result for 5 compounds.

Comp.	Log *p* ^a^	p*K*a ^a^	*M*_W_ ^a^	HBA ^a^	HBD ^a^	RB ^a^	SA ^a^	Electronegativity ^b^
Y12	2.11	5.7	355.34	6	2	7	342.81	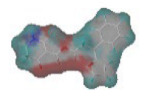
S1	2.66	5.6	389.79	6	2	7	361.58	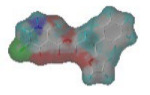
S2	2.57	5.6	389.39	6	2	7	362.88	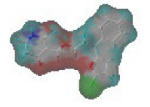
S3	1.49	5.6	370.36	7	3	7	358.47	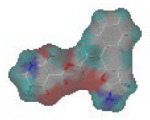
S4	2.86	5.6	434.24	6	2	7	370.40	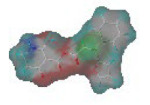
S5	2.86	5.6	434.24	6	2	7	368.19	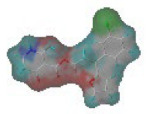

Notes: ^a^ DS for predicting Log *p*, p*K*a, molecular weight (*M*_W_), hydrogen bond acceptor (HBA) and donor (HBD), rotatable bonds (RB) and surface area (SA); ^b^ SYBYL for predicting electronegativity.

**Table 7 ijms-23-07822-t007:** Pyridine derivatives used for the Topomer CoMFA analysis.

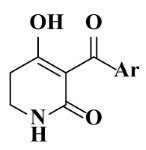				
Comp.	Ar	IC_50_ (μM)	pIC_50_	
		Obsd.	Obsd.	Pred.
ZD-1	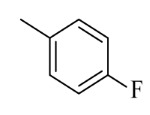	1.247	5.904	5.840
ZD-2 *	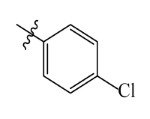	1.276	5.894	5.877
ZD-3	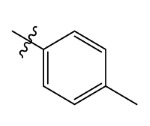	1.365	5.865	5.860
ZD-4	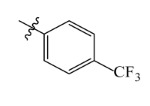	0.998	6.001	5.997
ZD-5	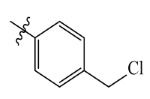	1.324	5.878	5.879
ZD-6	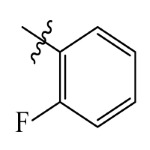	5.000	5.301	5.402
ZD-7	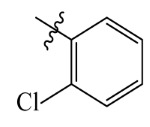	3.475	5.459	5.470
ZD-8 *	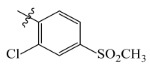	0.283	6.548	6.528
ZD-9	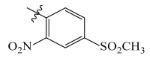	0.261	6.583	6.579
ZD-10	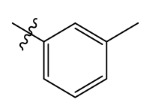	1.503	5.823	5.781
ZD-11	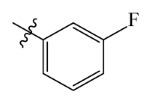	1.346	5.871	5.903
ZD-12 *	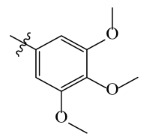	0.352	6.454	6.471
ZD-13	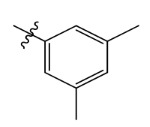	1.457	5.845	5.839
ZD-14	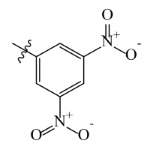	1.203	5.938	5.890
ZD-15	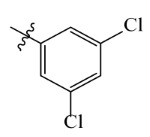	0.268	6.572	6.588
ZD-16 *	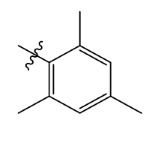	2.075	5.683	5.750
ZD-17	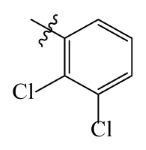	0.561	6.251	6.205
ZD-18	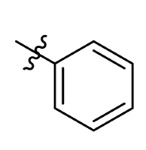	4.446	5.352	5.341
ZD-19	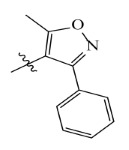	3.327	5.478	5.480
ZD-20	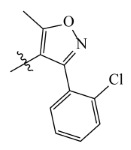	3.206	5.494	5.541
ZD-21 *	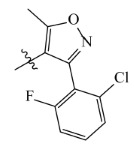	2.506	5.601	5.576
ZD-22	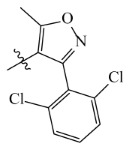	3.076	5.512	5.533
ZD-23	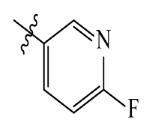	1.321	5.879	5.880
ZD-24	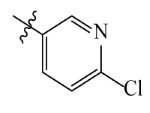	1.202	5.920	5.930
ZD-25	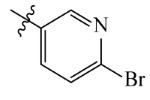	1.138	5.944	5.960
ZD-26	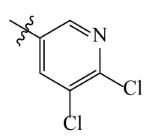	0.998	6.001	6.082
ZD-27	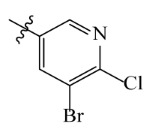	0.425	6.372	6.333
ZD-28	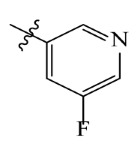	1.782	5.749	5.891
ZD-29 *	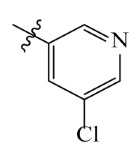	1.589	5.799	5.888
ZD-30	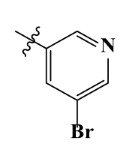	1.347	5.868	5.911
ZD-31	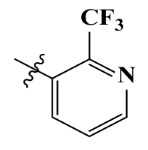	2.238	5.676	5.590
ZD-32	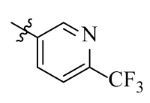	1.419	5.848	5.838
ZD-33 *	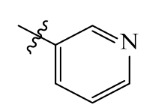	7.656	5.116	5.140
ZD-34	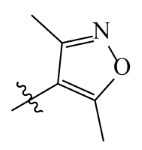	4.256	5.371	5.380
ZD-35	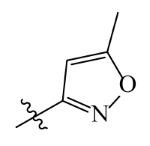	4.009	5.397	5.361

* Compounds were considered as the test set. IC_50_: Half maximal inhibitory concentration toward *At*HPPD.

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
