# Peer review of "Computer-Aided and AILDE Approaches to Design Novel 4-Hydroxyphenylpyruvate Dioxygenase Inhibitors"

_ijms, 2022, doi:10.3390/ijms23147822_

Round 1

Reviewer 1 Report

·         Revise the title as the colon is unnecessary

·         The dataset is very small therefore the results cannot be applied to the broad chemical space and also lack general applicability.

·         The huge gap between r2 and q2 indicates over-fitting, kindly justify.

·         Authors are advised to check the following sentence “Plants in the sun threaten life, eventually 32 leading to death with albino symptoms if HPPD inhibitors interfere with the transfor- 33 mation of HPPA to HGA”.

·         The word combination “HPPD herbicides” is incorrect; it should be “HPPD based herbicides”

·         The sentences like “It is the most economical way to employ computer technology in drug discovery”, etc. clearly indicate that the English throughout the manuscript needs improvements.

·         Introduction section is too short.

·         All abbreviations be explained on their first appearance.

·         The introduction lacks the previous work done in the field.

·         The author must clarify whether the molecules in the dataset are pyridine derivatives or ‘cyclic lactone’ derivatives.

·         The Figure 2 clearly indicates that the test and training set are not balanced. None of the high activity compound is in test set.

·         The claim that “the experimental and predicted values were uniformly distributed near the 45° line, which indicated excellent prediction ability.” requires adequate references and justification.

·         Authors are advised to provide images with better resolution.

·         The Figure 4 requires the inclusion of labels for all residues in the vicinity of native and docked ligand.

·         Authors must mention all the interatomic bonding distances for all ligands.

·         What is that gray colored ‘CO A:502‘ in Figure 5?

·         Authors are advised to discuss the interaction energy (CDOCKER energy) of ligands with respect to native ligand.

·         The MD requires clarification about inclusion of heavy atoms of ligands, as they have already given backbone Cα.

·         In MD simulation, author must give RMSF plot along with discussion on the radius of gyration.

·         What is the reason for selecting the pdb 6JX9?

Reviewer 2 Report

The manuscript “Design of Novel 4-Hydroxyphenylpyruvate Dioxygenase Inhibitors: Based on Computer-Aided and AILDE Approaches describes a workflow based on computer-aided methods, through which 5 candidate compounds for HPPD inhibition are proposed. The topic is of interest because inhibition of HPPD is a relatively new direction in pesticide research. However, in order to published the manuscript in the IJMS journal, the following issues need to be improved or clarified:

-        - Please revise the Introduction so that there is a smooth transition in the paragraphs that describe the methods used in the paper.

-        - What were the criteria for selecting the compounds in the training and test set?

-        - For the topomer COMFA step, the minimum energy conformation of the most active compound, ZD-9, was used. Would not its "active" conformation have been more appropriate? Where active refers to ZD-9 conformation docked at HPPD site.

-        In its current form, the work seems unfinished. It would be advisable for the authors to find a way to validate their candidate compounds.

Round 2

Reviewer 1 Report

The authors have revised the manuscript and addressed some issues, but the dataset used by them is small and confined to a single scaffold. Therefore, the work lacks general applicability and addresses small chemical space. In addition, some issues are not addressed properly.

1.      The revised title “Based on Computer-Aided and AILDE Approaches Designing of Novel 4-Hydroxyphenylpyruvate Dioxygenase Inhibitors” still requires some additional modifications. A suitable title could be “Computer-Aided and AILDE Approaches to design Novel 4-Hydroxyphenylpyruvate Dioxygenase Inhibitors”.

2.      In his seminal paper “Best Practices for QSAR Model Development, Validation,

and Exploitation” by Alexander Tropsha (DOI : 10.1002/minf.201000061), he has clearly suggested that the dataset should have at least 40 compounds, which is a reasonable number for performing QSAR analysis. The paper is based on a dataset of 34 molecules only.

3.      Figure 2 clearly indicates that the training set and test set are not equally distributed. Authors are advised to read the following paper from Dr. Dearden et al. SAR and QSAR in Environmental Research, Vol. 20, Nos. 3–4, April–June 2009, 241–266.

4.      It is really surprising that the authors are claiming “In the process of Topomer CoMFA model generation, the obtained r2 and q2 make the model have reliable predictive ability by repeatedly changing and constantly adjusting the molecules of training set.”. This clear indicates that there are good chances of information leakage. Authors are advised to read the seminal paper by Masand et al. (Med Chem Res (2015) 24:1241-1264

DOI: 10.1007/s00044-014-1193-8)

5.      The sentence “The results were verified by comparing the RMSD between the original ligand and the same interaction site as the original ligand.” is not clear. Where is RMSD value?

6.      The claim that the ligand after redocking of the original ligand (red) completely overlapped with the ligand in the complex (cyan) is not right. They don’t have 100% overlap. Hence, RMSD is must.

7.      Surprisingly, in previous version of the same manuscript (Figure 4), the ligand has different conformation especially for terminal rings. A strong explanation is required with adequate references for this anomaly.

8.      In all the molecules, authors could consider the existence of tautomerism.

9.      In Figure 5, distances for different types of interactions are missing.

Reviewer 2 Report

The authors have made significant improvements to the manuscript, but I do not consider that only proposing candidate compounds is a suitable conclusion for IJMS. Therefore, I do not recommend the manuscript publication in its current form.

Round 3

Reviewer 1 Report

No new comments

Reviewer 2 Report

The authors have made significant improvements to the manuscript, but I do
not consider that only proposing candidate compounds is a suitable conclusion for International Journal of Molecular Sciences. Therefore, I do not recommend the manuscript publication in its current form.
